# Therapeutic Approaches for C9ORF72-Related ALS: Current Strategies and Future Horizons

**DOI:** 10.3390/ijms26136268

**Published:** 2025-06-28

**Authors:** Marco Cattaneo, Eleonora Giagnorio, Giuseppe Lauria, Stefania Marcuzzo

**Affiliations:** 1Neuroalgology Unit, Fondazione IRCCS Istituto Neurologico Carlo Besta, 20133 Milan, Italy; marco.cattaneo@istituto-besta.it (M.C.); giuseppe.lauriapinter@istituto-besta.it (G.L.); 2PhD Program in Pharmacological Biomolecular Sciences, Experimental and Clinical, University of Milan, 20133 Milan, Italy; 3Neurology 4—Neuroimmunology and Neuromuscular Diseases, Fondazione IRCCS Istituto Neurologico Carlo Besta, 20133 Milan, Italy; 4Department of Medical Biotechnology and Translational Medicine, University of Milan, 20133 Milan, Italy; 5Brain-Targeted Nanotechnologies (BraiNs) Lab, Fondazione IRCCS Istituto Neurologico Carlo Besta, 20133 Milan, Italy

**Keywords:** amyotrophic lateral sclerosis, C9ORF72, small molecules, biological drugs, therapeutic strategies

## Abstract

Amyotrophic lateral sclerosis (ALS) is a fatal neurodegenerative disease characterized by the loss of upper and lower motor neurons. One of its major genetic causes is C9ORF72, where mutations lead to hexanucleotide repeat expansions in the C9ORF72 gene. These expansions drive disease progression through mechanisms, including the formation of toxic RNAs and the accumulation of damaged proteins such as dipeptide repeats (DPRs). This review highlights these pathogenic mechanisms, focusing on RNA foci formation and the accumulation of toxic DPRs, which contribute to neuronal damage. It also discusses promising targeted therapies, including small molecules and biological drugs, designed to counteract these specific molecular events. Small molecules such as G-quadruplex stabilizers, proteasome and autophagy modulators, and RNase-targeting chimeras show potential in reducing RNA foci and DPR accumulation. Furthermore, targeting enzymes involved in repeat-associated non-AUG (RAN) translation and nucleocytoplasmic transport, which are crucial for disease pathogenesis, opens new therapeutic avenues. Even some anti-viral drugs show encouraging results in preclinical studies. Biological drugs, such as antisense oligonucleotides and gene-editing technologies like CRISPR-Cas, were explored for their potential to specifically target C9ORF72 mutations and modify the disease’s molecular foundations. While preclinical and early clinical data show promise, challenges remain in optimizing delivery methods, ensuring long-term safety, and improving efficacy. This review concludes by emphasizing the importance of continued research and the potential for these therapies to alter the disease trajectory and improve patient outcomes.

## 1. Introduction

The pathogenic mechanisms associated with the C9ORF72 gene, particularly in the context of C9ORF72-linked amyotrophic lateral sclerosis (ALS), are complex and multifactorial, involving alterations in transcription, RNA foci formation, and the accumulation of toxic proteins such as dipeptide repeats (DPRs) [1]. These alterations contribute to neuronal dysfunction and degeneration, underscoring the urgent need to better understand the molecular underpinnings of the disease. Clinically, patients with ALS caused by C9ORF72 repeat expansions often present with motor impairment accompanied by cognitive decline and/or behavioral changes [2].

The development of targeted therapies is therefore essential, as they aim to directly address the molecular causes of the disease, with the goal of slowing or halting its progression. Among the most promising strategies are small molecules and biological drugs. While small molecules can be effective in certain contexts, they have limitations related to specificity and potential off-target side effects [3]. In contrast, biological drugs, particularly antisense oligonucleotides (ASOs), provide a more selective approach by directly targeting pathogenic RNA transcripts, thereby reducing the production of toxic DPR proteins [4].

Biological therapies represent an innovative and highly specific approach to targeting C9ORF72 mutations, with promising results in preclinical models and some clinical trials. However, long-term efficacy, safety, and delivery methods remain areas of ongoing development. By combining targeted approaches with advances in therapeutic technologies, new avenues may open for the treatment of C9ORF72-linked neurodegenerative diseases, with the potential to significantly improve the quality of life for patients.

This review offers a comprehensive overview of current and emerging therapeutic strategies for C9ORF72-linked ALS, with a focus on advances in small molecules and biological drugs. As research into the molecular basis of C9ORF72 pathology progresses, these targeted therapies hold substantial promise for transforming patient care and potentially altering the disease trajectory.

## 2. Physiological and Pathological Roles of C9ORF72 in ALS

### 2.1. C9ORF72 Physiological Functions

As shown in Figure 1, the C9ORF72 gene spans 41 kb and is located on chromosome 9, as its name suggests. It can be transcribed into three distinct mRNA isoforms through alternative splicing. Two of these isoforms encode the longer version of the protein, known as C9ORF72-Long (C9-L), which has a molecular weight of 54 kDa, while the third encodes the shorter isoform, C9ORF72-Short (C9-S), with a molecular weight of 24 kDa. C9ORF72 mRNA is expressed in most tissues, although it is least abundant in lymphoid tissues [5]. The C9-L isoform is primarily found in the cytoplasm, while the C9-S isoform is predominantly associated with the nuclear membrane. Although the precise function of each isoform remains unclear, C9-S is thought to play a role in regulating nuclear–cytoplasmic transport, as suggested by immunoprecipitation studies [6]. C9-L, on the other hand, interacts with various organelles, including the Golgi apparatus, stress granules, mitochondria, and lysosomes. Based on these interactions, it is believed that C9-L is involved in processes like axonal growth, mitochondrial oxidative phosphorylation, and autophagy [5,7,8]. Indeed, C9ORF72 shares structural homology with the Differentially Expressed in Normal and Neoplastic Cell (DENN) protein family, which functions as a regulator of Rab GTPases. These GTPases play a central role in autophagy and vesicle trafficking [9], highlighting the involvement of C9ORF72 in autophagic processes, intracellular transport, and key cellular functions such as mitochondrial dynamics and axonal growth [10].

### 2.2. Pathological Roles of C9ORF72 in ALS

Among the various genetic contributors to ALS, C9ORF72 mutations play a major role, accounting for up to 40% of familial and 10% of sporadic cases [11]. Clinically, these patients are notable for a high comorbidity with frontotemporal dementia (FTD), leading to the recognition of a distinct ALS-FTD subtype [12]. The pathology is driven by a hexanucleotide repeat expansion (GGGGCC, or G4C2) in the first intron of the gene. While healthy individuals typically carry 11 or fewer repeats, affected patients can have several hundred. Expansions over 30 repeats are considered pathogenic. Although this mutation does not alter the protein structure or disrupt translation, it still impairs cellular function [13].

The exact pathogenic mechanisms underlying C9ORF72-related ALS are not yet fully understood. Two main hypotheses have been proposed to explain the cellular toxicity, though neither has been definitively confirmed. The first suggests a toxic gain of function from the expansion-containing RNA. Since the repeat expansion lies within a non-coding region, the resulting C9ORF72 protein remains structurally unaffected. However, once transcribed and spliced out from the coding sequence, the repeat-rich RNA is prone to forming abnormal secondary structures due to its high guanine and cytosine content. In particular, it can intercalate into DNA and form R-loops—three-stranded RNA–DNA hybrid structures that interfere with transcription and compromise genome stability [1,10,14]. These structures are also able to sequester RNA-binding proteins and form aggregates, which are called RNA foci, affecting the posttranscriptional process [15]. They are found in both neurons and glial cells located in the motor and frontal cortex, hippocampus, cerebellum, and the spinal cord [16]. Moreover, the aberrant RNA is able to exit the nucleus, and once in the cytosol, it causes ribosomes to initiate translation without a start codon. This leads to the production of small proteins made up of dipeptide repetitions (DPRs): poly-GA, poly-GP, poly-GR, poly-PA, poly-GP, and poly-PR [17]. The toxicity is developed from the formation of aggregates and inclusions, the disruption of nucleocytoplasmic transport and protein translation, and the impairment of protein homeostasis [18].

The second toxic mechanism involves a loss of function of the C9 proteins due to decreased expression. It has been shown that the expansion binds to trimethylated histones, leading to a reduction in gene transcription, which is further diminished by the hypermethylation of CpG islands in the promoter region [19]. These two different processes cause a reduction in C9 expression and, therefore, an insufficient amount of protein to maintain cellular homeostasis. The exact mechanism behind toxicity remains debated, making the development of effective therapies more complex, as all hypotheses must be considered.

An additional challenge is the absence of fully reliable animal models for C9ORF72-ALS. Several mouse models have been developed, each capturing a specific aspect of the disease. Knockout models are mainly used for loss-of-function studies [20,21,22], while BAC transgenic mice are employed for gain-of-function research [21,23]. However, these models vary slightly, and none fully replicate the entire clinical picture. Drosophila models are limited to gain-of-function studies [24] due to the absence of a C9ORF72 orthologue, while zebrafish models can simulate various mechanisms, though not all at once [25]. Lastly, C. Elegans models can replicate several processes, as demonstrated in a study by Sanobe and colleagues, which showed that reducing poly-GA and poly-GP levels extends the lifespan of C. elegans models [26]. However, several cellular and molecular processes remain underexplored.

## 3. Small Molecules as Therapeutic Strategies for C9ORF72-Associated ALS

Small molecules represent a promising class of therapeutic compounds in ALS. Their key advantage lies in their ability to cross biological barriers, ensuring good bioavailability even in challenging targets, without the need for complex delivery systems. From an industrial perspective, this makes production more straightforward and cost-effective [27]. Over the years, various processes have been explored as potential targets for small-molecule-based therapies.

### 3.1. Small Molecules Targeting RNA Structure

One of the toxicities from C9ORF72-ALS stems from the formation of stable secondary structures in the expanded RNA called G-quadruplexes [28], which can hybridize with DNA to form R-loops (RNA∙DNA hybrids), and single-strand hairpins [29]. These structures can sequester and mislocalize specific proteins involved in the splicing and nuclear transport processes [29] while also altering transcription processes [30]. Su and colleagues approached the problem by first noting a structural similarity between loops found in the fragile-X syndrome (poly-CGG) and those in ALS. Molecules that are able to bind FRAXA expansion are already known, and they have been confirmed to also bind C9ORF72 expansion. They then demonstrated how molecules capable of binding G-quadruplexes can inhibit Ran translation and foci formation in neurons derived from fibroblasts carrying the C9ORF72 mutation [31]. Building on this, a more recent study identified a small molecule that selectively bound r(G4C2)exp, prevented the sequestration of RNA-binding proteins, and inhibited RAN translation [32].

Bush and colleagues developed a dual-function small molecule that selectively binds the three-dimensional structure formed by r(G4C2) exp and recruits an endogenous ribonuclease (RNase) to cleave it, an approach known as an RNase-targeting chimera (RIBOTAC). RIBOTACs are designed to engage RNA quality control pathways, promoting the degradation of toxic RNAs and alleviating associated cellular defects. The authors showed that their RIBOTAC effectively cleaves r(G4C2) exp, rescuing C9ORF72 HRE-associated pathologies in multiple models, including iPSC-derived spinal neurons and a C9ALS/FTD BAC transgenic mouse. Transcriptome-wide analyses confirmed its selectivity for the expanded repeat in intron 1 of C9ORF72 [33].

### 3.2. Small Molecules Acting on DPR Accumulation

Regarding DPR accumulation, Simone and colleagues demonstrated the effectiveness of the RIBOTAC strategy using structurally distinct molecules. Using G-quadruplex stabilizers, they demonstrated that DPRs and RNA foci burden are reduced after treatment, both in iPSC-derived neurons and mutation-carrying Drosophila models, leading to a significant increase in survival [29]. In another study, researchers showed that geldanamycin (GELD) and spironolactone (SPL) effectively reduce DPR levels through distinct mechanisms. GELD inhibits Hsp90 and promotes proteasomal degradation, while SPL, an aldosterone antagonist, enhances autophagy. The study also found that general translation inhibitors similarly decrease DPR production and that the modulation of the cAMP-PKA pathway directly influences DPR levels. These findings open up several promising avenues for the development of new therapeutic strategies [34].

An alternative therapeutic strategy focuses on targeting the enzymes involved in the translation of the repeat expansion that generates DPRs via RAN translation. This process is regulated by the double-stranded RNA-dependent protein kinase (PKR), and compounds that modulate PKR activity may effectively reduce DPR production [35]. Zu and colleagues investigated the therapeutic potential of metformin, a drug that is widely used in the treatment of diabetes, based on prior evidence showing its ability to suppress RAN translation in a mouse model of Huntington’s disease. In their study, metformin was shown to reduce DPR translation in HEK cells without altering C9ORF72 protein levels. Additionally, metformin decreased the levels of phosphorylated PKR, consistent with reduced PKR activity. These findings were further validated in the bacterial artificial chromosome (BAC) C9 mouse model, showing an improvement in motor parameters such as brake, brake/stance, and brake/stride and anxiety-like behavior [36]. Of note, based on this evidence, the University of Florida started a phase 1 clinical study using metformin in ALS patients, with an estimated completion date in 2025 (ID: NCT04220021).

Building on these findings, Hatanaka and colleagues investigated the effects of rifampicin on DPR accumulation and demonstrated that intranasal administration reduced both DPR levels and phospho-TDP43 in BAC mice. They further confirmed that this effect was associated with the modulation of PKR phosphorylation. Notably, rifampicin also reduced RNA foci, which are structures that are not directly linked to PKR activity, indicating the involvement of an additional PKR-independent mechanism. Moreover, the treated mice showed an increase in memory acquisition and retention compared to untreated animals. However, the authors highlight the need for further investigation, particularly regarding drug formulation, as long-term rifampicin treatment has been associated with liver toxicity [37].

### 3.3. Small Molecules for the Restoration of Nucleocytoplasmic Transport

The impact of C9ORF72 in nucleocytoplasmic transport is still understudied in the context of pharmacological therapy. This type of active transport requires Ran GTPase to hydrolyze GTP to GDP, and the Ran guanine nucleotide exchange factor (RanGEF) is used to refurbish the used GTP. Ran–GTP is present in the nucleus, while Ran–GDP is in the cytosol, and GDP functions as a signal to import it back into the nucleus [38]. Zhang and colleagues demonstrated that hexanucleotide repeats may alter this transport, causing Ran to accumulate in the cytosol. Treating a Drosophila model with the quadruplex-binding porphyrin compound, TMPyP4 5,10,15,20-Tetrakis-(N-methyl-4-pyridyl) porphine rescues both import defects and neurodegeneration [38]. An intriguing prospect is to target proteins involved in nuclear transport, as one of the defects in C9-mutated cells involves nucleocytoplasmic transport, particularly an increase in nuclear export. An experiment carried out by Zhang and colleagues showed how the inhibition of nuclear export or the enhancement of nuclear import can prevent C9ORF72 neurodegeneration [38].

### 3.4. Antiviral Drugs for C9ORF72-ALS

In a similar framework, antiviral drugs may offer promising potential. Human endogenous retroviruses (HERVs) are retroviruses that have integrated in the human genome in the last 5 million years, and they represent ~8% of the human genome. They can be reactivated by environmental stimuli, and they are linked to various neurological disorders. One of such endogenous retroviruses, HERV-K, plays a role in human development but is silenced in the later stages of fetal life. However, if reactivated in neurons, HERV-K causes cell death [39]. Increasing evidence links HERV-K and ALS, including the presence of reverse transcriptase activity in CSF and blood, as well as an increase in HERV-K transcripts in the brains of ALS patients. Despite this, several antiviral monotherapies have been tested without success. However, combination antiretroviral therapy (ART), such as the one used to treat HIV, proved promising. Notably, the use of Triumeq (a combination of dolutegravir, abacavir, and lamivudine) reduced both the neurophysiological index’s decline and slowed the progression of ALSFRS-R in a phase 2a trial [40]. A phase 3 trial is currently underway, with expected completion in 2026 (ID: NCT05193994). Another retrotransposable element (RTE) that has been linked to neurological disorders is the long interspersed elements (LINEs). LINEs are non-long terminal repeat RTEs that contribute to genetic diversity, even in diseases. LINE-1, which constitutes 17% of the human genome, is particularly successful due to its ability to encode machinery for its own mobilization. LINE-1 retrotransposition is highly active in human neural progenitor cells, and its expression is higher in the brain compared to other organs. LINE-1 is linked to various neurodegenerative diseases, including ALS, where it is associated with TDP-43 proteinopathy [41]. Specifically, Liu et al. showed that TDP-43 normally suppresses retrotransposition, and its loss may contribute to neurotoxicity, particularly DNA damage and neuroinflammation [42]. Furthermore, LINE-1 products were found to be elevated in C9ORF72-mutated ALS patients [43]. As a result, a phase 2a study using censavudine, the most potent known LINE-1 transcriptase inhibitor, has just concluded with promising results, and a phase 3 trial is currently in planning (ID: NCT04993755).

**Figure 1 ijms-26-06268-f001:**
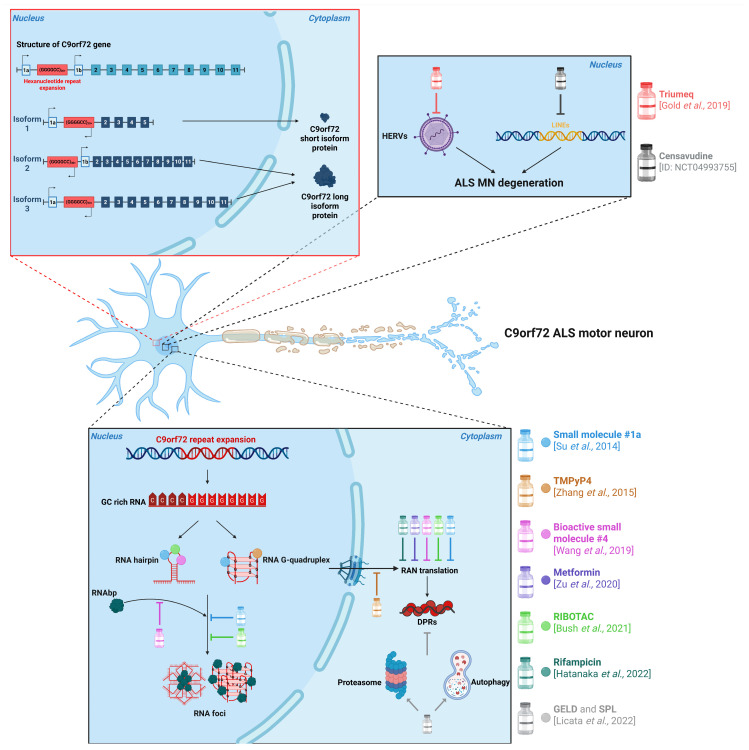
Overview of C9ORF72 gene structure and small molecules as therapeutic strategies for C9ORF72-associated ALS. Left upper panel: Schematic representation of the structure of the C9ORF72 gene with exons (numbered blue squares) and the G_4_C_2_ repeat expansion highlighted in red. Below, the three transcribed isoforms (Isoform 1, Isoform 2, and Isoform 3) are shown, along with their corresponding translated proteins. Right upper panel: HERVs (purple) and LINEs (yellow) are both linked to ALS neurodegeneration. On the right, the main antiviral drugs tested in ALS, with the inhibition lines showing the drug effects. Lower panel: HRE (red) in the C9ORF72 gene (blue) is transcribed, and it can form different structures, including G-quadruplexes and RNA hairpins. These RNA structures can sequester RNA-binding proteins (RNAbp), thus forming RNA foci in the nucleus. If G-quadruplexes and RNA hairpins are translocated outside the nucleus, they are translated via repeat-associated non-ATG (RAN) translation, in which different dipeptide repeat proteins (DPRs) can be formed. The panel on the right shows the small molecules tested as therapeutic compounds in C9ORF-ALS [31,32,33,34,36,37,38,40]. The balls show the binding sites, whereas the inhibition lines show the drug effects on the molecular and biological pathways of C9ORF-ALS motor neurons. Figure created with Biorender.com (Biorender, Toronto, ON, Canada): Cattaneo, M. (2025) https://BioRender.com/1h88uu6 (accessed on 23 March 2025).

## 4. Biological Drugs for Treating C9ORF72-Associated ALS

### 4.1. RNA-Targeted Therapies

As shown in Figure 2, biological drugs represent a promising avenue in the treatment of ALS, offering the ability to target specific molecular mechanisms with greater precision and fewer off-target effects compared to small molecules. One of the most immediate ways to impact C9ORF72 pathology is through ASOs [4]. They are small nucleic sequences, which are complementary to a sequence of a specific mRNA, designed to block its translation and cause its degradation by cellular machinery. Being nucleic acids, they can have a plethora of modifications to increase specificity, stability, and activity. They are already approved as therapies for SMA, Duchenne’s muscular dystrophy, and even SOD1-ALS [44]. Due to C9ORF-ALS having a clear genetic target, it is a promising strategy. In a single patient with a mutant C9ORF72 gene harboring the G4C2 repeat expansion, the repeated intrathecal dosing of the optimal ASO was well tolerated, resulting in significant reductions in cerebrospinal fluid poly(GP) levels. This report offers valuable insights into how nucleic acid chemistry influences toxicity, and to our knowledge, it is the first to demonstrate the feasibility of clinically suppressing the C9ORF72 gene. Further clinical trials are necessary to assess the safety and efficacy of this therapeutic approach in patients with C9ORF72 mutations [45]. Other studies have focused on mechanisms similar to ASOs, such as the RNAi-RISC complex. This approach uses either a miRNA or a single-hairpin RNA (shRNA) as a guide for cleavage by the Argonaute enzyme [46]. Martier and colleagues refined this strategy, culminating in the development of concatenated artificial miRNA hairpins targeting the G4C2 expansion. These hairpins reduce C9ORF72 transcript in the nucleus and cytoplasm, together with RNA foci in iPSC-derived neurons and mouse models [47,48]. Another group employing an artificial miRNA targeting C9ORF72 noted a reduction in both C9ORF72 mRNA and DPRs; they, however, noted that a more promising approach should be to target the expansion exclusively [49].

### 4.2. Protein-Targeted Therapies

A hallmark of C9ORF72-associated pathology is the disruption of the autophagic process [50]. Shi and colleagues explored the effects of an engineered anticoagulation-deficient form of activated protein C (3K3A-APC) in iPSC-derived motor neurons from ALS patients. They found that treatment with this engineered protein reduced the burden of DPRs, corrected autophagosome abnormalities, and increased survival. Furthermore, they discovered that this protective effect was mediated by the activation of the coagulation factor II thrombin receptor (F2R-PAR1) pathway [51].

### 4.3. Gene Editing

Gene editing represents one of the most promising avenues in the treatment of C9ORF72-linked ALS. Among the various approaches, the CRISPR-Cas system has emerged as the most extensively studied. However, a significant limitation lies in the fact that the key components of this system, Cas proteins and guide RNAs, are not naturally expressed in human cells. The most straightforward strategy involves transfecting affected cells with plasmids encoding the CRISPR-Cas machinery. Although this method poses challenges for clinical translation, several preclinical studies have demonstrated its potential, showing reductions in both RNA foci and DPR levels [52,53].

Building on these findings, more recent studies have focused on improving the delivery of the CRISPR-Cas system. Meijboom and colleagues used an adeno-associated viral vector (AAV) to deliver the system. However, because the CRISPR-Cas9 apparatus is large, they required two separate vectors. Despite this, they successfully demonstrated the excision of the C9ORF72 expansion, which led to reduced RNA foci, lowered DPR synthesis, and increased C9ORF72 protein expression in both iPSCs and BAC mice [54]. However, the authors raised concerns about the safety profile of their delivery method, citing potential risks such as genomic instability due to continuous nuclease activity, AAV genome integration, and an increased immune response triggered by the bacterial Cas9 protein. As a result, they suggested exploring less time-stable delivery methods. To address these concerns while still utilizing the advantages of AAV vectors, Cox and colleagues replaced Cas9 with a variant of Cas13, known as CasRx. Unlike Cas9, which targets DNA, Cas13 targets RNA [55]. This modification has the potential to reduce risks of genomic instability or insertion, as it does not interact with the DNA. However, Cas13 requires longer expression to exert therapeutic effects, which is where AAV vectors’ long-term expression capacity becomes valuable. Additionally, CasRx is smaller than Cas9, allowing it to be packaged into a single vector. The authors demonstrated a reduction in the C9ORF72 repeat-containing transcripts and observed less neurotoxicity in both iPSCs and animal models. In the zebrafish model, this approach rescued larval hyperactivity. Nonetheless, they noted several critical points, including the potential for immune responses due to prolonged Cas expression, no significant impact on DPR levels, and low transduction efficiency in vivo [56]. In conclusion, while the use of biological drugs is still in its early stages of development, these treatments hold significant promise, offering the potential to slow disease progression and improve patient outcomes.

**Figure 2 ijms-26-06268-f002:**
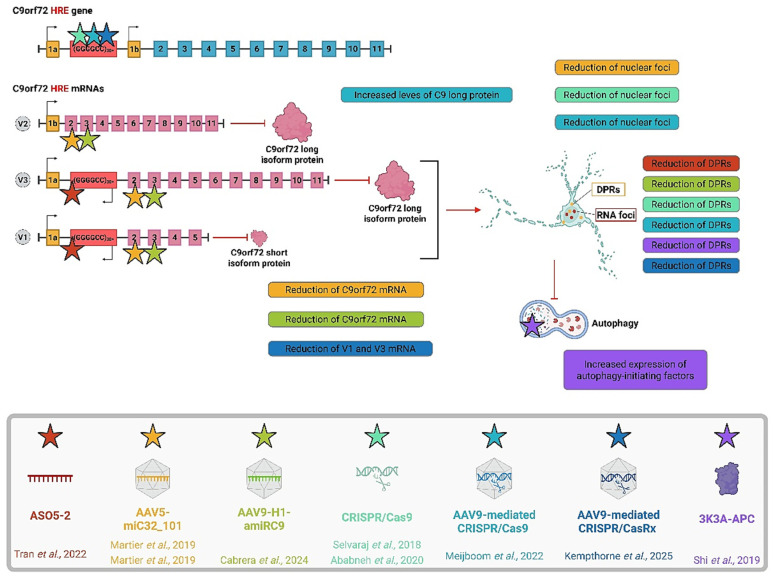
Overview of biological drugs for treating C9ORF72-associated ALS. The expanded C9ORF72 human locus includes two non-coding exons (yellow square, 1a and 1b) and 10 coding exons (blue squares, from 2 to 11), with the G4C2 repeat (red) abnormally expanded and localized between the two non-coding exons. C9ORF72 gene gives rise to three coding variants (grey circles): variant 1 (V1), which includes exon 1a and exons 2–5; variant 2 (V2), which includes exon 1b and exons 2–11; and variant 3 (V3), which includes exon 1a and exons 2–11. Alternative splicing of these three RNA variants results in the production of the C9 protein’s (pink cloud) short isoform and the C9 protein’s long isoform. The coding exons gene (blue squares), mRNAs (pink squares), and non-coding exons (yellow squares). Abnormally expanded G4C2 repeat (red) causes the bidirectional transcription of the HRE, generating sense and antisense expanded RNAs. These HRE transcripts give rise to G-quadruplex and hairpin structures that can form RNA foci and, if translated, DPRs that, in turn, cause autophagy impairments. In the grey box, the main biological drugs tested in vitro or in vivo for treating C9ORF-ALS are indicated [45,47,49,51,52,54,56]. The stars indicate the target sequence of the relative drug (color-matched). In the colorful boxes, the effect of the respective drug (color-matched) is shown. Figure created with Biorender.com (Biorender, Toronto, ON, Canada): Cattaneo, M. (2025) https://BioRender.com/1h88uu6 (accessed on 23 March 2025).

## 5. Conclusions

As discussed here, the complex pathogenic cellular and molecular mechanisms underlying C9ORF72-linked ALS present significant challenges for effective therapeutic intervention. However, as our understanding of these molecular processes deepens, targeted therapies are emerging as promising avenues for treatment. Both small molecules and biological drugs offer new strategies that precisely address the root causes of the disease. While small molecules face limitations in specificity and off-target effects, biological therapies provide greater precision and have shown encouraging results in preclinical studies and some clinical trials. Despite these advancements, key issues related to long-term efficacy, safety, and delivery still need to be fully resolved. Moving forward, combining innovative therapeutic technologies with targeted approaches will likely open new opportunities for treating C9ORF72-linked ALS, offering hope for improved patient outcomes and potentially altering the progression of this devastating disease.

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
