# Peer review of "Therapeutic Approaches for C9ORF72-Related ALS: Current Strategies and Future Horizons"

_ijms, 2025, doi:10.3390/ijms26136268_

Round 1

Reviewer 1 Report

Comments and Suggestions for Authors

This review summarized the pathogenic mechanisms of C9ORF72 in ALS and introduced therapeutic strategies for C9ORF72-Associated ALS including small molecular and biological drugs. Overall this review provides the recent advances and might help to better understand the theraputical researches of C9ORF72-Associated ALS.

Minor comments:

1.A figure showing the structure of C9ORF72 might help to understand, including the information about different isoforms and hexanucleotide repeat expansion.

2.Clinical symptoms of C9ORF72-Associated ALS should be briefly described. And it would be useful to mention whether any of the discussed therapeutic strategies have shown efficacy in alleviating these symptoms.

Author Response

Referee # 1:

This review summarized the pathogenic mechanisms of C9ORF72 in ALS and introduced therapeutic strategies for C9ORF72-Associated ALS including small molecular and biological drugs. Overall this review provides the recent advances and might help to better understand the theraputical researches of C9ORF72-Associated ALS.

Minor comments:

1.A figure showing the structure of C9ORF72 might help to understand, including the information about different isoforms and hexanucleotide repeat expansion.

We thank the Reviewer for this insightful and constructive comment. Following your suggestion, we have carefully revised the figure 1 including the molecular information of C9orf72 gene

2.Clinical symptoms of C9ORF72-Associated ALS should be briefly described. And it would be useful to mention whether any of the discussed therapeutic strategies have shown efficacy in alleviating these symptoms.

We wish to thank the Reviewer for this helpful comment that allows us to revise the manuscript and discuss the clinical symptoms of C9orf72-associated ALS in the Introduction, 3.2 and 4.3 paragraphs.

Reviewer 2 Report

Comments and Suggestions for Authors

This is an excellent review of the complexities of Amyotrophic Lateral Sclerosis (ALS) that arises from mutations in the C9ORF72 gene. This review is important for, among other reasons, mutations in C9OF72 are now believed to account for ~20% of mono-genetic ALS and ~40% of "sporadic" ALS. The authors have provided a lucid explanation of the varying functions and known consequences of expanded introns 1 (with repeated G4C2) that remain in the nucleus, and various mRNA's that escape the nucleus. Normals (non-ALS) have 0-30 expansions and ALS subjects can have hundreds to thousands of these expansions in intron 1 of C9ORF72.

The therapeutic implications of this knowledge have led to two general approaches and are also discussed fairly. Some approaches appear better than others in terms of altering the course of ALS, and this review should stimulate development of additional approaches.

The included two Figures are excellent as teaching tools and contribute to the paper.

English is fine, needs slight editing.

Author Response

Referee # 2:

This is an excellent review of the complexities of Amyotrophic Lateral Sclerosis (ALS) that arises from mutations in the C9ORF72 gene. This review is important for, among other reasons, mutations in C9OF72 are now believed to account for ~20% of mono-genetic ALS and ~40% of "sporadic" ALS. The authors have provided a lucid explanation of the varying functions and known consequences of expanded introns 1 (with repeated G4C2) that remain in the nucleus, and various mRNA's that escape the nucleus. Normals (non-ALS) have 0-30 expansions and ALS subjects can have hundreds to thousands of these expansions in intron 1 of C9ORF72.

The therapeutic implications of this knowledge have led to two general approaches and are also discussed fairly. Some approaches appear better than others in terms of altering the course of ALS, and this review should stimulate development of additional approaches.

The included two Figures are excellent as teaching tools and contribute to the paper.

English is fine, needs slight editing.

We thank the reviewer for their kind and encouraging comments. We have revised the manuscript to improve the editing particularly in the Introduction.